# STREET: A Multi-Task Structured Reasoning and Explanation Benchmark

**Danilo Ribeiro**[1,2,*]**, Shen Wang**[1,†]**, Xiaofei Ma**[1]**, Henghui Zhu**[1]**, Rui Dong**[1]**, Deguang Kong**[1]**,
Juliette Burger**[1]**, Anjelica Ramos**[1]**, William Wang**[1]**, Zhiheng Huang**[1]**, George Karypis**[1]**,
Bing Xiang**[1]**, Dan Roth**[1]

[1] AWS AI Labs,
{shenwa, xiaofeim, henghui, ruidong, kongdegu, burgerju}@amazon.com,
{anjeramm, wyw, zhiheng, gkarypis, bxiang, drot}@amazon.com
[2] Northwestern University,
{dnribeiro}@u.northwestern.edu,

## Abstract

We introduce STREET, a unified multi-task and multi-domain natural language reasoning and explanation benchmark. Unlike most existing question-answering (QA) datasets, we expect models to not only answer questions, but also produce step-by-step structured explanations describing how premises in the question are used to produce intermediate conclusions that can prove the correctness of a certain answer. We perform extensive evaluation with popular language models such as few-shot prompting GPT-3 and fine-tuned T5. We find that these models still lag behind human performance when producing such structured reasoning steps. We believe this work will provide a way for the community to better train and test systems on multi-step reasoning and explanations in natural language.

## 1 Introduction

A long-term pursuit in Artificial Intelligence is to endow machines with the ability to reason and manipulate premises to reach conclusions and perform tasks. Initially, most reasoning systems performed multi-step operations over symbolic or probabilistic knowledge (Newell & Simon, 1956; McCarthy et al., 1960; Siler & Buckley, 2005), and even though these systems were able to perform complex tasks (Vernon et al., 2007; Metaxiotis et al., 2002; Ribeiro & Forbus, 2021), there were still shortcomings when it comes to encoding such knowledge, learning reasoning rules and dealing with ambiguity (Bell, 1985; Ribeiro et al., 2019). Some recent works in the field of *question-answering* (QA) have demonstrated that language models can bypass some of these issues and learn to reason directly over natural language (Clark et al., 2020), allowing for more flexible and adaptable reasoning capabilities. Another advantage of performing multi-step reasoning over natural language is that it allows for more inspectable outputs, improving the explainability of models that are otherwise regarded as black box systems (Jain & Wallace, 2019; Rajani et al., 2019a; Danilevsky et al., 2020). Despite the recent progress, we notice that there is still a gap in resources for training and evaluating general reasoning capabilities over natural language.

To facilitate research in this direction we propose the *STructured REasoning and Explanation Multi-Task* benchmark (or STREET for short), containing a collection of tasks in various domains including quantitative reasoning (math questions), analytical reasoning (logic puzzle questions), and deductive reasoning (common-sense and science questions). We build upon existing QA datasets by adding multi-premise, multi-step, structured explanations in the form of *reasoning graphs*, as depicted in Figure 1. The STREET benchmark contains 35.8k questions, each of which is accompanied by a reasoning graph, either created by expert annotators or programmatically. When combined, all reasoning graphs contain a total of 151.1k reasoning steps (or textual entailments), of which 14.7k

---

[*] Work done during internship at AWS AI, current address: Department of Computer Science, Northwestern University, Evanston, IL 60208, USA

[†] Corresponding Author.

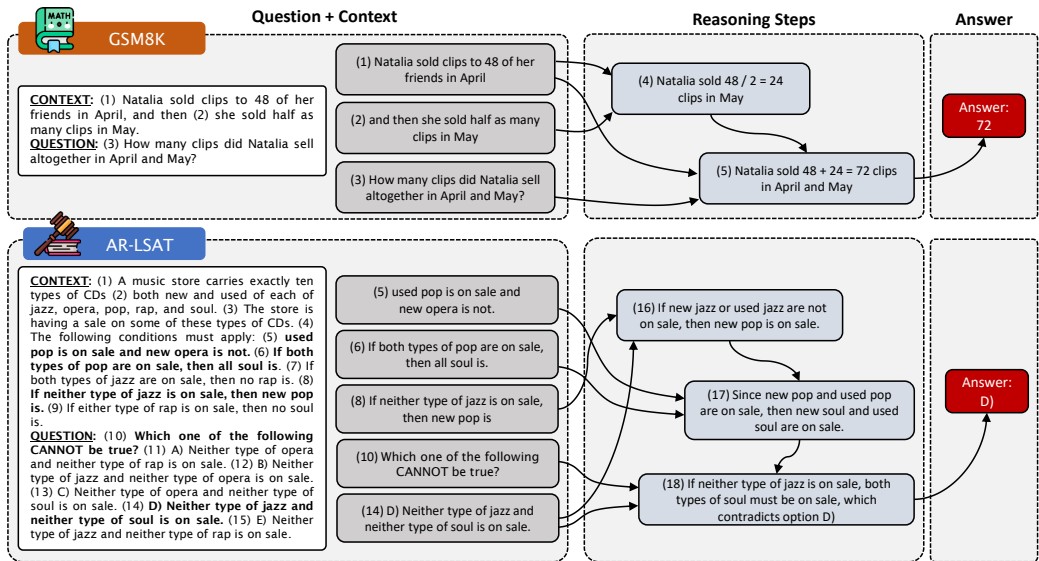

Figure 1: Two examples from our proposed STREET benchmark. The questions are derived from the Grade School Math (GSM8K) and Analytical Reasoning - Law School Admission Test (AR-LSAT) tasks. The QA components (e.g. question, context, and answers options) are broken into *textual logical units*, or TLUs. These TLUs are connected to form a *reasoning graph*. Our proposed benchmark builds upon existing QA datasets by adding structured reasoning explanations that shows how one can derive the answer to a given question.

were created by our expert annotators. We carefully selected the tasks such that most of the relevant knowledge required to answer the questions is contained within the question or context themselves.

Therefore, we focus on the reasoning problem, with a greater number of reasoning steps (an average of 7.8 reasoning steps per answer) and a more complex reasoning structure than previous datasets. These properties differentiate our work from single-step reasoning such as Natural Language Inference (NLI) (Bowman et al., 2015; Williams et al., 2018; Zellers et al., 2018) or multi-hop QA (Yang et al., 2018; Chen et al., 2021) that require specific factual knowledge retrieval.

In our proposed evaluation, the models are expected to not only answer the questions, but also generate the reasoning graphs (including the textual intermediate steps) that explains their output answer. With that in mind, we design a few evaluation metrics to verify if the generated reasoning graphs match the expected golden data. We perform extensive evaluation using some popular language models of various sizes, namely T5 (Raffel et al., 2020) and GPT-3 (Brown et al., 2020), either fine-tuning on training data or using few-shot prompting. Our experiments show that even though these models can achieve high solving rates on many of the original QA datasets, they still struggle to generate coherent and relevant reasoning graphs and appear to be far below human performance.

Our main contributions are as follows: (1) We define reasoning graphs, which are structured chains of reasoning in natural language that provide explainability to the output of models on QA tasks. (2) We propose STREET, a multi-task and multi-domain benchmark containing questions requiring diverse types of reasoning skills. The answers in the dataset contain annotated or generated reasoning graphs. We make the data and evaluation code available online [1] (3) We evaluate the performance of LMs such as fine-tuned T5 and few-shot prompting with GPT-3 on our proposed task. Our results suggest there is still room for improving language models when it comes to generating complex multi-step reasoning explanations.

---

[1]https://github.com/amazon-science/street-reasoning

## 2 TASK AND DATA

### 2.1 TASK DEFINITION

In the standard definition, a question-answering task $\boldsymbol{T} = (C, Q, O, A, R)$ has the following components: an optional context $C$ such as a passage or problem description; a question $Q$ that might reference the context; answer options $O = (o_1, o_2, \ldots, o_K)$ in the case of $K$-way multiple choice questions; an expected answer $A$ (where $A \in O$, if options are present). Some MRC tasks (Ling et al., 2017; Camburu et al., 2018; Rajani et al., 2019b; Cobbe et al., 2021) also provide rationales $R$ as free-form textual explanations of the answer $A$.

To generate a more fine-grained explanation, we modify the above formulation so that the data also contains a structured, step-by-step explanation of the answer, as depicted in Figure 1. To this end, we define **textual logical units** (TLU), which are essentially a sequence of tokens from elements in $\boldsymbol{T}$ that defines premises that will possibly be referenced in the reasoning steps. More formally, a TLU for a QA component in $T \in \boldsymbol{T}$ is a list of the sequence of tokens in $T$. For instance, given the tokens $T = (t_1, t_2, \ldots, t_{|T|})$, the TLU of $T$ is defined as the list of spans $L_T = ((l_1, l_2), (l_2 + 1, l_3), \ldots, (l_{(|L_T|-1)} + 1, l_{|L_T|}))$ where $l_i \in \{x \mid 1 \leq x \leq |T|\}$ and $l_i > l_j$ for $i > j$. Each pair $(l_i, l_j) \in L_T$ represents the sequence of tokens $(t_{(l_i)}, t_{(l_i+1)}, \ldots, t_{(l_j)})$. The TLUs can be extracted automatically from the text by using a simple algorithm, e.g., breaking down paragraphs by punctuation mark. The algorithm we used to create the datasets can be found in Appendix A.5. Note that the TLUs for the rationale $L_R$ can be thought of as a sequence of step-by-step explanations.

Given the TLUs, we also define a structured explanation, which we call **reasoning graph**. Each element in $L_R$ (referred here as reasoning steps or intermediate conclusions) will be connected to a set of other TLUs (or premises) that contain sufficient information supporting the reasoning step. The reasoning graph can be defined as a set of vertices and edges $\mathcal{G} = (\mathcal{V}, \mathcal{E})$, where the nodes are TLUs such that $\mathcal{V} \subseteq (L_C \cup L_Q \cup L_O \cup L_A \cup L_R)$ and the edges $\mathcal{E} \subseteq \mathcal{V} \times (L_O \cup L_A \cup L_R)$ are pairs of nodes. The starting node of an edge is a premise of a reasoning step, while the ending node is the output of a reasoning step, i.e., an intermediate conclusion or an answer. Note that in this case the explanation graph $\mathcal{G}$ is **directed** (edges go from premises to conclusions) and **acyclic** (each conclusion should only be generated once). Our reasoning graph formulation shares some similarities to Entailment Trees from Dalvi et al. (2021). However, our benchmark does not require a pre-assigned corpus of textual facts or a hypothesis (which must be included with the data). Furthermore, reasoning graphs allow for other QA elements (e.g., context, answer options, and expected answer) and represents the reasoning using the less restrictive directed acyclic graphs (a tree data structure can't easily be used to represent the examples from Figure 1).

### 2.2 DATA SOURCE AND ANNOTATION

With the goal of testing complex reasoning capabilities, we build upon existing QA datasets in which solutions require multiple reasoning steps. Since our focus is testing reasoning capabilities, we disregard datasets that require domain knowledge to solve the problem. Instead, we focus on the ones containing most of the information within the context, question, and answer options. We categorize the reasoning tasks according to their level of existing structured reasoning steps, which we describe below.

The first category, comprised of the science question dataset AI2 Reasoning Challenge (ARC) (Clark et al., 2018), already contains annotated structured reasoning steps provided by ENTAILMENTBANK (Dalvi et al., 2021). ENTAILMENTBANK comes with an external source of knowledge (Jansen et al., 2018) from which premises could be retrieved to generate explanations. Since the retrieval of premises is out of the scope of this work, we directly add the gold premises to the context $C$ of the QA task[2].

The second category uses the Sequential Context-Dependent Execution dataset (SCONE) (Long et al., 2016). The questions in SCONE describe a sequence of actions that modify a given toy world (e.g., list of objects and their relative position), and the expected answer is the final world state. We extract

---

[2] Entailment Bank has a similar formulation which they call Task-1. However, they do not consider the problem of choosing the correct answer given the multiple choice options from ARC.

| Task Name | Task Domain | # Original Questions | # Used Questions | # Reasoning Steps | Answer Type |
|-----------|-------------|----------------------|------------------|-------------------|-------------|
| ARC | Science | 7,787 | 1,840 | 5,881 | 4-Way MC |
| SCONE | Processes | 14,574 | 14,574 | 130,482 | State Pred. |
| GSM8K | Math | 8,792 | 1,030 | 4,666 | Number |
| AQUA-RAT | Math | 101,449 | 1,152 | 7,179 | 5-Way MC |
| AR-LSAT | Logic | 2,046 | 500 | 2,885 | 5-Way MC |
| TOTAL | — | 134,648 | 19,096 | 151,093 | — |

Table 1: The different tasks used to create the proposed benchmark. In the answer types, "$K$-Way MC" stands for multiple choice answer with $K$ options.

the reasoning steps programmatically as this dataset also provide the intermediate world states for each question.

The third category of tasks, namely GSM8K (Cobbe et al., 2021) and AQUA-RAT (Ling et al., 2017) contain **unstructured** rationales (in natural language) showing the chain of reasoning used to solve the given questions. In this category, we further annotate the datasets. First, we split the context, question, answer options, and rationales into TLUs, assigning a number ID to each of them. Afterwards, we ask human annotators to create the structure of the reasoning steps, assigning which premises were used in each entailment step. Note that some entailment steps might not have any premises (e.g., when stating a known fact as in "one hour has 60 minutes").

Finally, the last category is comprised of the AR-LSAT dataset (Zhong et al., 2021), which is a relatively complex reasoning task (transformer-based models are shown to struggle in this task) and does not come with any rationale or explanation. Therefore, we annotate the rationales and reasoning structure from scratch. We ask expert annotators to solve the questions given the context and the answer. While writing down the step-by-step explanations, they are also asked to assign which premises are used to reach each intermediate conclusion.

## 2.3 DATASET DETAILS

A summary of all datasets in our proposed benchmark can be found in Table 1. Note that not all questions in the original datasets were used due to the significant amount of time required for annotation. Examples of the final data for each dataset can be found in Appendix A.1.

**Annotation Details:** Each dataset is pre-processed and QA components are broken down into TLUs. For all datasets except ARC and SCONE, we ask human annotators to further annotate data points by first creating multi-step rationales (this is skipped if the dataset already contains rationales), and then connect each rationale step to the premises that support that conclusion (in the user-interface the annotators select a set of numbers for each rationale step). Note that human annotators are given an unlimited amount of time to complete each task, and they are mostly comprised of experts with undergraduate or graduate level education, as opposed to randomly selected online workers.

For quality control, we performed two passes for each reasoning step, using a third pass to break ties when needed. As an indicator of annotation quality, we compute the annotation agreement using Fleiss Kappa $\kappa$ (Fleiss, 1971). Each directed edge in the reasoning graph is regarded as binary question (edge should be present or not). Finally, the first two annotation passes are used to compute $\kappa = 0.79$, indicating "substantial agreement" among annotators. With a total of 14,730 reasoning steps annotated (for GSM8K, AQUA-RAT, and AR-LSAT), we estimate a total of 1,467 (paid) work hours. Further annotation details can be found in Appendix A.2.

**Data Statistics:** We analyze the data of all combined datasets to obtain some insights into the scale and reasoning complexity. Figure 2 shows the distribution of the "number of reasoning steps" among the data points in each annotated dataset. Note that most of the tasks contain a larger number of reasoning steps compared to previous multi-hop QA datasets (Jhamtani & Clark, 2020; Yang et al., 2018; Geva et al., 2021; Chen et al., 2021). For the most part, multi-hop QA questions only contain up to two "hops" (or reasoning steps), while STREET has an average of 7.8 steps per question, with

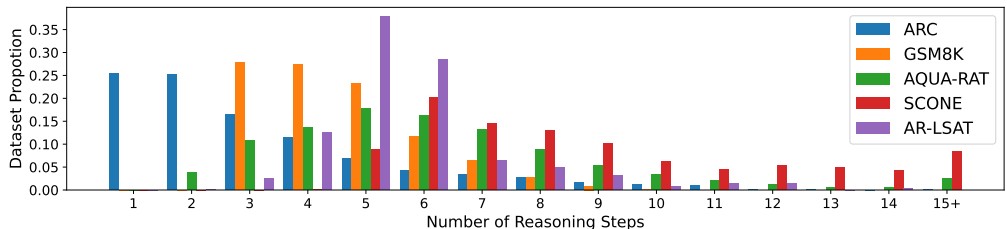

Figure 2: A histogram containing the distribution of data points and the number of reasoning steps for each annotated dataset (training, development, and testing split combined), truncated to a maximum of 15 steps. The distribution varies among datasets, with an average of 7.8 steps across all data points.

26.4% of the questions containing more than 10 reasoning steps. The number of incoming edges for each node (a.k.a "in-degree" or "valency", which is the number of directed edges with such node as destination) in the reasoning graph is usually between one and three, with questions containing nodes with more than five incoming edges. Further data statistics and dataset analysis are available in Appendices A.3 and A.4.

## 3 BASELINE MODELS

We measure the performance of various models on our multi-task reasoning and explanation benchmark. We show results separately for each of the tasks, where we mainly evaluate (1) the standard QA accuracy (i.e., can the model predict the correct answer?) and (2) the models' ability to generate the reasoning graph. The evaluation metrics will be described in section 4.1.

To this end, we use two approaches to solve the structured reasoning task. The first approach is fully supervised, where we fine-tune a T5 model (Raffel et al., 2020), which is similar to other work on reasoning datasets (Tafjord et al., 2021; Dalvi et al., 2021; Ribeiro & Forbus, 2021). The second approach uses the much larger GPT-3 language model (Brown et al., 2020). Instead of fine-tuning GPT-3, we use few-shot prompting. These large language models have been shown to have strong step-by-step reasoning generation capabilities (Wei et al., 2022; Wang et al., 2022) even if just provided with a handful of examples.

### 3.1 REASONING GRAPH ENCODING

Following prior work with structured input and output (Chen et al., 2020; Tafjord et al., 2021; Dalvi et al., 2021; Neves Ribeiro et al., 2022), we linearize the reasoning graph such that it can be generated by the language model as a sequence of tokens. First, each one of the candidate premises (i.e., context, question, and answer options) are assigned an id token. Then we sort the reasoning graph's steps according to a valid topological order (i.e., all premises must be part of the linearized string before adding a reasoning step node). For tasks where answer types are multiple-choice or number, the last node will contain the text with the value of the predicted answer, such as "The answer is A)" or "The answer is 15". The text encoding for the GSM8K example in Figure 1 can be seen below:

```
$question$ = (1) Natalia sold clips to 48 of her friends in April,
and then (2) she sold half as many clips in May. (3) How many
clips did Natalia sell altogether in April and May?

$proof$ = (1) & (2) -> (4): Natalia sold 48/2 = 24 clips in May;
(1) & (3) & (4) -> (5): Natalia sold 48+24 = 72 clips altogether
in April and May; (3) & (5) -> (6): The answer is 72;
```

The SCONE task is a special case where we do not expect the generative models to output the state of every tracked object in the answer node. Instead, the answer is extracted from the intermediate nodes of the reasoning graph (examples are shown in Appendix A.1)

## 3.2 SUPERVISED TRAINING

For full supervision, we fine-tune the T5-large model (770 million parameters) on the training data for each task separately. The model is fine-tuned for up to 30 epochs, and we select the check-point with the highest answer accuracy on the development data at the end of each training epoch. The training is done using a machine with four NVIDIA Tesla V100-SXM2, and the Hugging Face[3] pre-trained T5-model distribution. Further implementation details are available in Appendix C. During inference, we use beam search with a beam size of 5 to generate the reasoning graph and the answer for a given question.

## 3.3 FEW-SHOT PROMPTING

For few-shot prompting we use GPT-3 (Brown et al., 2020) by accessing the OpenAI's API [4]. The API provides access to a few model variants. For our experiments we use the largest advertised model, namely `text-davinci-002` (175B parameters). During inference, we select up to 5 examples (depending on the tasks and models, fewer prompt examples might be provided due to the encoder token size limit) as prompts for the model, following the encoding method from Section 3.1, except we remove the expected reasoning graph from the target question. During generation, we use greedy decoding and do not set any maximum or minimum output size, expecting the model to predict the end of the structured output.

# 4 EXPERIMENTS

## 4.1 EVALUATION METRICS

We specify the 3 main categories of evaluation metrics described below to evaluate the answer to the questions and the generated reasoning graph.

**Answer Accuracy:** This metric measures the ability of generative models to predict the correct answer to a question. Exact match is used to evaluate the models on tasks with *multi-choice* or *numerical* answers. For the tasks with state prediction (i.e., SCONE), we use the combined state for each object as the expected answer. The answer accuracy will be an upper bound for the other metrics since any generated reasoning graph with an incorrect answer is also labeled as incorrect.

**Reasoning Graph Accuracy:** This metric compares the predicted and golden reasoning graphs in terms of both the graph structure and the content of the intermediate conclusion nodes. The comparison between the predicted graph $\mathcal{G}_p = (\mathcal{V}_p, \mathcal{E}_p)$ and golden graph $\mathcal{G}_g = (\mathcal{V}_g, \mathcal{E}_g)$ starts with aligning the nodes in $\mathcal{V}_p$ and $\mathcal{V}_g$. In this matching, we use the premises as anchors, and the reasoning step nodes are matched according to their ancestors in a topological ordering. Given two matched reasoning step nodes $v_p \in \mathcal{V}_p$ and $v_g \in \mathcal{V}_g$, we use textual similarity function $\sigma(text(v_p), text(v_g))$ to test if two reasoning step nodes are equivalent. The textual similarity function varies depending on the QA Task. More details on the matching algorithm and the different text similarity functions used are available in Appendix D. Note that this is a strict metric, and small deviations from the golden reasoning graph will render the predicted graph incorrect.

**Reasoning Graph Similarity:** The reasoning graph similarity $sim(\mathcal{G}_p, \mathcal{G}_g)$ is a "softer" metric that compares the predicted and golden reasoning graphs using the graph edit distance function $\delta(\mathcal{G}_p, \mathcal{G}_g)$. The function $\delta$ uses *insertion*, *deletion* and *substitution* as elementary graph edit operators over nodes and edges. The text similarity function $\sigma$ is used to test if two nodes match. The cost of any edit operation is $1$. However, if the generated answer is incorrect, the similarity is set to 0 (i.e., the edit cost for the "answer node" $L_A$ is $\infty$). The Graph Similarity function is normalized (the output is in the range $[0, 1]$) and can be computed as:

$$sim(\mathcal{G}_p, \mathcal{G}_g) = 1 - \left[ \frac{\delta(\mathcal{G}_p, \mathcal{G}_g)}{max(|N_p| + |E_p|, |N_g| + |E_g|)} \right] \tag{1}$$

---

[3]model available at `https://huggingface.co/t5-large`
[4]https://openai.com/api/

| Model | ARC | SCONE | GSM8K | AQUA-RAT | AR-LSAT |
|---|---|---|---|---|---|
| **Answer Accuracy** | | | | | |
| Random Guess | 25.0 | — | — | 20.0 | 20.0 |
| T5 [large] (fine-tuned) | 93.5 | 69.6 | 10.4 | 28.7 | 28.0 |
| GPT-3 [davinci] (few-shot) | 72.9 | 02.3 | 34.8 | 40.2 | 19.0 |
| **Reasoning Graph Accuracy** | | | | | |
| T5 [large] (fine-tuned) | 17.1 | 60.0 | 00.7 | 00.0 | 00.0 |
| GPT-3 [davinci] (few-shot) | 01.7 | 01.2 | 00.7 | 00.0 | 00.0 |
| **Graph Similarity** | | | | | |
| T5 [large] (fine-tuned) | 44.1 | 67.0 | 05.4 | 00.9 | 00.3 |
| GPT-3 [davinci] (few-shot) | 15.1 | 01.9 | 16.0 | 05.2 | 01.1 |

Table 2: The main results on the test set across the different tasks and different evaluation metrics. Numbers are in percentage. The "Random Guess" results are included to facilitate visualization since different tasks have different answer types.

In general, computing the graph edit distance can be computationally expensive as the problem is *NP-complete* (Abu-Aisheh et al., 2015). For this reason, we compute an approximation of this value by using the implementation from the networkx[5] library.

## 4.2 RESULTS

The main experiment results can be found in Table 2. The results for the SCONE task are averaged across the different sub-tasks (namely Alchemy, Scene, and Tangrams). Further experiment results with different model sizes and generalization settings can be found in Appendix B.

There are a few takeaways. First, we notice that T5 [large] (fine-tuned) either outperforms or is on par with GPT-3 [davinci] (few-shot) on ARC and SCONE across all metrics, while the opposite is true for the math domain tasks GSM8K and AQUA-RAT. Both methods perform poorly on AR-LSAT. This result is consistent with the results from Zhong et al. (2021), which shows that language models still struggle with more complex analytical reasoning tasks.

Second, the results of fine-tuned T5 [large] show that the Reasoning Graph Accuracy and Reasoning Graph Similarity are substantially better on both ARC and SCONE (with perfect generation for 17.1% and 60.0% of the correct answers, respectively), while close to zero for the remaining tasks. Both ARC and SCONE are the tasks with best answer accuracy compared to "random guess". This suggests a trend that the higher the answer accuracy for a task, the more likely the model is able to produce a correct reasoning graph explaining their answer.

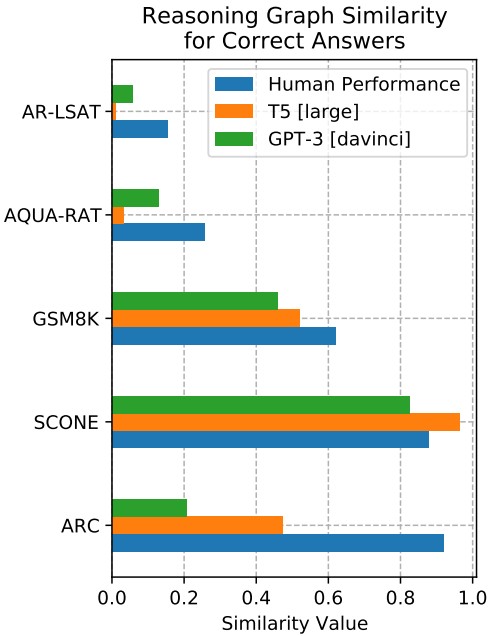

Figure 3: A histogram containing the reasoning graph similarity of baseline models as well as human performance on a randomly selected subset of the test data.

[5]https://networkx.org/

Third, the GPT-3 [`davinci`] model struggles
with tasks outside of the math domain. Noticeably, the SCONE results are far worse when compared
to T5 [`large`]. This has been observed previously by Wei et al. (2022), where few-shot prompting
models can struggle with toy "symbolic reasoning" tasks. This often happens when models are not of
sufficient size or if the prompt examples are out of domain (i.e., evaluation examples have more steps
than prompt examples).

To better visualize and understand the quality of the generated output, we plot the reasoning graph
similarity for each task in Figure 3. This plot only considers outputs in which answers match the
golden answers for the QA tasks. For reference, we also estimate the human performance by asking
expert annotators to write the reasoning graph from scratch, given only the context, question, and
the expected answer (broken down into TLUs) for 100 randomly selected questions from the test set
across the different tasks.

The baseline models perform well compared to human performance on both SCONE and GSM8K.
Most noticeably, T5 [`large`] seems to generate the best structured explanations outputs for SCONE,
which is expected since this task is more formulaic and contains the most training examples. On the
other hand, the baseline models seem to perform much worse on ARC, AQUA-RAT and AR-LSAT,
with human-generated reasoning graphs having over 200% higher scores compared to baselines. The
human results on AQUA-RAT, and AR-LSAT are somewhat lower than on the other tasks, primarily
due to the diversity of the possible valid outputs (there are multiple ways one can explain an answer).
In general, automatic evaluation of generated text is still a challenge (Celikyilmaz et al., 2020) in the
field of natural language understanding.

### 4.2.1 ERROR ANALYSIS

To better understand the model's mistakes, we manually analyze 100 randomly selected outputs
where the answers are incorrect. Since each STREET task has their own peculiarities, we analyze the
error patterns for each of them separately.

**ARC**    In this task, both baseline models seem to struggle with large outputs (i.e., more than 5
reasoning steps), which leads to a common error pattern where generated reasoning graphs do not
contain the answer to the question ($\approx 62\%$).

**SCONE**    To our surprise, GPT-3 [`davinci`] often fails to execute basic operations in this task. It
generates an incorrect conclusion in first reasoning step for $\approx 83\%$ of the analyzed outputs. This
could be due to the limited number of few-shot examples (as a consequence of the input size limit) or
because this task is outside of its pre-training data distribution. On the other hand, the T5 [`large`]
seems to make fewer mistakes, with $\approx 74\%$ of all reasoning steps matching the golden data.

**GSM8K**    Even among incorrect answers, both models are often able to output partially correct
proofs. A small percentage of the incorrect steps are due to calculation error ($\approx 12\%$ in GPT-3
[`davinci`] outputs and $\approx 30\%$ in T5 [`large`] outputs). In $\approx 37\%$ of the generated outputs, the
models seem to have misunderstood the question or applied the wrong formula in one of the steps.

**AQUA-RAT**    In this task, both models hallucinate the predicted answer into the last reasoning step
($\approx 28\%$), even when it does not follow the step's equation. Similarly to GSM8K, a small percentage
of the incorrect steps are due to calculation errors ($\approx 12\%$).

**AR-LSAT**    Both models struggle with this task. Most of the correct answers are due to random
chance, and ($\approx 33\%$) of the generated outputs don't contain an answer to the question. In particular,
GPT-3 [`davinci`] often just copies the TLUs from the question without making any meaningful
conclusions.

## 5    RELATED WORK

**Complex Reasoning in Question-Answering**    Modeling complex reasoning is an important chal-
lenge and a crucial component of natural language understanding. In the context of question-
answering (QA), initial efforts to emulate complex reasoning used symbolic representations and

problem solvers (Forbus & De Kleer, 1993; Platonov et al., 2020). With recent advances in pre-trained language models, reasoning over natural language has became more tractable as these models can more easily handle ambiguity and learn the reasoning rules implicitly (Clark et al., 2018; Tafjord et al., 2021). As one of the simpler forms of reasoning, textual entailment was extensively studied, and many datasets and tasks have been proposed (Bowman et al., 2015; Williams et al., 2018; Zellers et al., 2018). To address multi-step reasoning over language, many multi-hop QA (MHQA) datasets and methods were proposed (Yang et al., 2018; Dua et al., 2019; Xiong et al., 2021; Chen et al., 2021). Common limitations of MHQA that we try to address in this paper include (1) a small number of reasoning steps (usually up to three) and (2) simplistic evaluation, allowing models to correctly predict answers by exploiting spurious correlations (Tang et al., 2021). Datasets such as CLUTRR (Sinha et al., 2019) and RuleTaker D* (Clark et al., 2020) better addressed the multi-step and structured aspect reasoning with explanations. However, they contain mostly synthetic data and tasks that are relatively easy to solve with current language models.

**Explainable Question-Answering**   Due to the black-box nature of neural networks (Danilevsky et al., 2020), many approaches were proposed to improve the explainability of QA systems. They include explanation graphs (Jansen et al., 2018), generating a free-form natural language explanations (Camburu et al., 2018; Rajani et al., 2019b; Ayyubi et al., 2020), and chain of reasoning explanations Jhamtani & Clark (2020). Most noticeably, Dalvi et al. (2021) introduced the concept of *Entailment Trees*, containing multi-premise textual entailment steps. Entailment Trees differ from our proposed Reasoning Graphs in three main points (1) they were designed to be used mostly for explanation, not representing the answer to the questions directly (2) they require an external corpus of premises (3) they use the concept of hypothesis (as a combination of question + answer) that needs to be annotated as input. We believe reasoning graphs are a more flexible representation for explaining answers in the context of QA.

**Multi-Task Language Understanding**   Benchmarks are an important way to measure progress in natural language understanding. Datasets that contain multiple tasks have the advantage of testing the generality of models. The GLUE (Wang et al., 2018) and SUPER-GLUE (Wang et al., 2019) contain tasks such as reading comprehension and natural language inference. The Massive Multitask Language Understanding benchmark (Hendrycks et al., 2021) contains various QA problems extracted from the internet. BIG-Bench (Srivastava et al., 2022) contains over 200 tasks drawing on problems involving linguistics, math, common-sense reasoning, and others. These datasets arguably test the model's ability to perform reasoning to some degree. However, most of their evaluation revolves around answering questions instead of systematically testing the reasoning required to answer such questions.

## 6   CONCLUSION

We aim to enable machines to perform multi-step reasoning while explaining their answers. We believe that teaching machines how to manipulate premises and reach conclusions can be an important step towards true language understanding. With that in mind, we introduce STREET, a new multi-task reasoning and explanation resource covering various forms of reasoning in the context of question-answering. We hope this benchmark will allow for a more systematic evaluation of the reasoning capabilities of natural language systems. Future avenues of research include exploring reasoning capabilities with knowledge retrieval and using supervised models trained on multi-step reasoning data to bootstrap unsupervised learning for multi-step reasoning.

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

## A STREET DATA

### A.1 DATA EXAMPLES

We show examples (as linearized textual encoding) for each of the five tasks in STREET below. The questions and reasoning graphs were taken from the development set. The SCONE task contains three different sub-tasks (namely Alchemy, Scene, and Tangrams), and we show examples for each of them separately as their format differs significantly.

---

**ARC:**

```
$question$ = (1) the sun rising / setting occurs once per day
(2) the sun setting is a kind of event (3) the sun rising is a
kind of event (4) Which event occurs on a daily cycle? (5) A) The
Sun rises and sets. (6) B) Earth tilts on its axis. (7) C) Earth
revolves around the Sun. (8) D) The Moon revolves around Earth.

$proof$ = (1) & (2) & (3) -> (9): the sun rising and setting is
the event that occurs once per day; (9) -> (10): The answer is A);
```

---

**SCONE (Alchemy):**

```
$question$ = (1) first beaker has 0 chemicals (2) second beaker
has 1 green chemical (3) third beaker has 1 purple chemical (4)
fourth beaker has 1 orange chemical (5) fifth beaker has 1 green
chemical (6) sixth beaker has 1 red chemical (7) seventh beaker
has 1 yellow chemical (8) throw out the orange chemical (9) then,
add the leftmost beaker of green chemical to the yellow chemical
(10) mix it (11) then, add the remaining green chemical to it (12)
mix that too

$proof$ = (4) & (8) -> (13): fourth beaker has 0 chemicals; (2)
& (7) & (9) -> (14): seventh beaker has 1 yellow and 1 green
chemical; (2) & (9) -> (15): second beaker has 0 chemicals; (14)
& (10) -> (16): seventh beaker has 2 brown chemicals; (16) & (11)
& (5) -> (17): seventh beaker has 2 brown and 1 green chemicals;
(11) & (5) -> (18): fifth beaker has 0 chemicals; (17) & (12) ->
(19): seventh beaker has 3 brown chemicals;
```

---

**SCONE (Scene):**

```
$question$ = (1) position 1 has no person (2) position 2 has no
person (3) position 3 has no person (4) position 4 has no person
(5) position 5 has person in red shirt and yellow hat (6) position
6 has no person (7) position 7 has no person (8) position 8 has
no person (9) position 9 has no person (10) position 10 has no
person (11) a man in a yellow shirt appears on the right of the
man in a red shirt and yellow hat (12) a second man in a yellow
shirt appears on the left end (13) he leaves (14) the man in the
red shirt and yellow hat moves one space to the left (15) a man in
a red shirt appears on his right

$proof$ = (11) & (6) -> (16): position 6 has person in yellow
shirt and no hat; (1) & (12) -> (17): position 1 has person in
yellow shirt and no hat; (17) & (13) -> (18): position 1 has
no person; (14) & (4) & (5) -> (19): position 4 has person in
```

red shirt and yellow hat; (14) & (5) -> (20): position 5 has no
person; (20) & (15) -> (21): position 5 has person in red shirt
and no hat;

---

**SCONE (Tangrams):**

$question$ = (1) position 1 has figure A (2) position 2 has figure
D (3) position 3 has figure E (4) position 4 has figure C (5)
position 5 has figure B (6) swap the 1st and 5th figure (7) swap
the 1st and 3rd figure (8) swap them back (9) delete the 5th
figure (10) add it back

$proof$ = (1) & (6) -> (11): position 1 has figure B; (5) & (6) ->
(12): position 5 has figure A; (11) & (7) -> (13): position 1 has
figure E; (3) & (7) -> (14): position 3 has figure B; (13) & (8)
-> (15): position 1 has figure B; (14) & (8) -> (16): position 3
has figure E; (12) & (9) -> (17): position 5 has no figure; (17) &
(10) -> (18): position 5 has figure A;

---

**GSM8K**

$question$ = (1) Adam and Tom are brothers. (2) Adam is 8 years
old and (3) Tom is 12 years old. (4) In how many years will their
combined age be 44 years old?

$proof$ = (2) & (3) -> (5): At present, the two brothers have a
combined age of 8 + 12 = 20 years old.; (5) -> (6): Therefore,
1 year means an increase in the sum of their ages by 1 * 2 = 2
years.; (4) & (5) -> (7): Adam and Tom need a total of 44 - 20
= 24 more years to be 44 years old together.; (6) & (7) -> (8):
So both brothers will be 44 years old together after 24 / 2 = 12
years.; (4) & (8) -> (9): The answer is 12;

---

**AQUA-RAT**

$question$ = (1) Three birds are flying at a fast rate of 900
kilometers per hour. (2) What is their speed in miles per minute?
(3) [1km = 0.6 miles] (4) A)32400 (5) B)6000 (6) C)600 (7) D)60000
(8) E)10

$proof$ = (0) -> (9): To calculate the equivalent of miles in a
kilometer; (3) -> (10): 0.6 kilometers = 1 mile; (10) & (1) ->
(11): 900 kilometers = (0.6)*900 = 540 miles; (0) -> (12): In
1 hour there are 60 minutes; (11) & (12) & (2) -> (13): Speed
in miles/minutes = 60 * 540 = 32400; (13) & (2) & (4) -> (14):
Correct answer - A; (14) -> (15): The answer is A);

---

**AR-LSAT**

$question$ = (1) Four boys - (2) Fred, Juan, Marc, and Paul -
(3) and three girls - (4) Nita, Rachel, and Trisha - (5) will
be assigned to a row of five adjacent lockers, (6) numbered
consecutively 1 through 5, (7) arranged along a straight wall.
(8) The following conditions govern the assignment of lockers to
the seven children: (9) Each locker must be assigned to either one

Figure 4: Screenshot of the web-based annotation tool designed to author the reasoning graphs. The fields on the left contain the TLUs from the question components. The fields on the right are used by annotators to select the premises for the current rationale (i.e., reasoning step). For AR-LSAT, the annotators also had to write the rationales in addition to selecting the premises.

```
or two children, (10) and each child must be assigned to exactly
one locker. (11) Each shared locker must be assigned to one girl
and one boy. (12) Juan must share a locker, (13) but Rachel cannot
share a locker. (14) Nita's locker cannot be adjacent to Trisha's
locker. (15) Fred must be assigned to locker 3. (16) Which one of
the following is a complete and (17) accurate list of the children
who must be among those assigned to shared lockers? (18) A) Fred,
Juan (19) B) Juan, Paul (20) C) Juan, Marc, Paul (21) D) Juan,
Marc, Trisha (22) E) Juan, Nita, Trisha

$proof$ = (1) & (3) & (5) -> (23): Four boys and three girls will
be assigned to five adjacent lockers; (10) & (11) & (9) -> (24):
Each locker can be assigned to max two children, one girl and one
boy, and one child must be assigned to exactly one locker; (12)
& (14) -> (25): Kids who can share lockers : Juan, Nita, Trisha;
(13) -> (26): Kids not sharing lockers : Rachel; (0) -> (27):
Answer is 22; (27) -> (28): The answer is E);
```

## A.2 FURTHER DATA ANNOTATION DETAILS

Initially the expert annotators are given a guideline document containing a list of instructions on how to properly label the data. Afterwards, we carried out a meeting to elucidate any further questions about proposed annotation tasks. The guideline documents contains a few annotation examples created by the authors and also contains overall annotation instructions, summarized as follows:

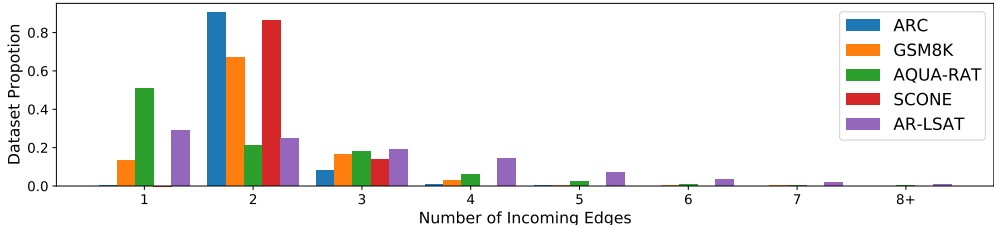

Figure 5: A histogram containing the distribution of incoming edges (a.k.a node's degree) per reasoning step for each annotated dataset (training, development, and testing split combined), truncated to a maximum value of 8 incoming edges.

- **Completeness:** The premises (directed edges) should contain all the information needed to ensure that the conclusion nodes are entailed given the premise.
- **Relevance:** Edges connecting nodes should ensure the entailment is correct, and no further irrelevant edges should be included.
- **Purposefulness:** The nodes containing the question and the answer should always be included in the final reasoning graph.
- **Granularity:** While writing the step-by-step rationales (in the case of AR-LSAT), the entailments should be fine-grained, encoding a single inference or logical step.

As mentioned previously, human annotators were given an unlimited amount of time to complete their tasks. The education level of the experts annotators were undergraduate or graduate. We believe all these factors contributed to the final quality of the data.

The annotation tool used a simple web-based form, with fields containing the question, context, answer choice, and answer. Each annotation represented a single entailment step. The annotators had to select from a check-box like input field containing premises. For AR-LSAT they were required to write the step rationales as well (explaining the answer). A screenshot of the annotation tool for with a GSM8K example is shown in Figure 4.

### A.3 FURTHER DATA STATISTICS

The STREET reasoning tasks are not only multi-step, but also multi-premise, where over 96.5% of nodes in the reasoning graphs contain two or more incoming edges. The distribution of such incoming edges is shown in Figure 5.

### A.4 DATASET ANALYSIS

We analyse the types inferences used during the reasoning steps in STREET. We sampled 100 reasoning steps, evenly distributed among the five tasks. We manually classify the reasoning steps in seven different categories, as shown in Table 3. For each category, we provide one reasoning step example, including the premises and the conclusion.

The majority of the reasoning steps requires "Spatial reasoning" (41%), involving concepts such as inertia, object displacement, ordering, and others. The second most common type involves "arithmetic inference" (20%), where conclusions contain simple operations such as addition, subtraction, multiplication and division. In the third most common inference type, "identifying answer" (17%), the conclusion simply states the final answer, either in a numerical or multiple-choice format.

### A.5 TEXTUAL LOGICAL UNIT EXTRACTION ALGORITHM

Algorithm 1 contains a pseudo-code for the script used to extract TLUs from the components of the questions. The algorithm uses Python's module "`re`" style of regular expression and matching to determine boundaries between TLUs.

| Inference Type | % | | Example(s) |
|---|---|---|---|
| spatial reasoning | 41 | *p1* | "position 3 has person in purple shirt and purple hat" |
| | | *p2* | "the man with the purple shirt and hat moves to the right end" |
| | | *step* | "position 3 has no person" |
| arithmetic inference | 20 | *p1* | "There are 30 students" |
| | | *p2* | "he wants to give a Valentine to 60% of them" |
| | | *step* | "He needs 18 valentine's cards because 30 * 0.6 = 18" |
| identifying answer | 17 | *p1* | "What is their speed in miles per minute?" |
| | | *p2* | "A) 32400" |
| | | *p3* | "Speed in miles/minutes = 60 * 540 = 32400;" |
| | | *step* | "The answer is A)" |
| logical inference | 12 | *p1* | "a plant cell is a kind of cell" |
| | | *p2* | "a cell nucleus is a part of a cell" |
| | | *step* | "a plant cell contains a nucleus" |
| paraphrasing question or context | 5 | *p1* | "Four boys" |
| | | *p2* | "and three girls" |
| | | *p3* | "will be assigned to a row of five adjacent lockers" |
| | | *step* | "Four boys and three girls will be assigned to five adjacent lockers" |
| algebraic manipulation | 4 | *p1* | "1/13 = 7/k" |
| | | *step* | "k = 91" |
| stating common knowledge | 1 | *step* | "In 1 hour there are 60 minutes" |

Table 3: Types of inference in the reasoning steps for the different tasks in STREET. *p1* through *p3* represent the premises, while *step* represents the conclusion. Some of the conclusions do not have any premises (e.g., "stating common knowledge" inference type).

## B  FURTHER RESULTS

We perform further experiments to understand: (1) how model sizes influence the final results (2) if models can generalize when using examples from other tasks. The experiment results with GPT-3 model are summarized in Table 4.

First, we compare the results with the GPT-3 [curie] (few-shot) model. This model performs significantly worse than GPT-3 [davinci] (few-shot), even though it is advertised as the next best model [6]. This phenomenon has been previously observed, with Wei et al. (2022) reporting that large models perform qualitatively better than smaller models. Second, the GPT-3 [davinci] (few-shot, MT) model selects one examples from each STREET task and use them as "few-shot" examples in order to simulate a multi-task setting. We observe that the results are comparable to GPT-3 [davinci] (few-shot) except for GSM8K, which has significantly worse results. For the most part, these results suggest that GPT-3 can adapt and learn from other reasoning domains.

---

[6]https://beta.openai.com/docs/models/gpt-3

---

**Algorithm 1** Textual Logical Unit Extraction

---

1: **procedure** EXTRACT_TLUS($text$)
2:     $patterns \leftarrow$ "$(.\ )|(,\ )|(!\ )|(?\ )|(\ and\ )|(\ then\ )$"              ▷ TLU boundaries REGEX
3:     $matches \leftarrow text.regex\_match(patterns)$
4:     $tlus \leftarrow []$
5:     $last\_pos \leftarrow 0$
6:     **for** $match \in matches$ **do**
7:         $tlu \leftarrow text.substring(last\_pos, match.end())$
8:         **if** $match.text() \in [",", "and", "then"]$ **then**
9:             **if** $tokens(tlu).size() \geq 5$ **then**
10:                 $tlus.push(tlu)$
11:                 $last\_pos \leftarrow match.end$
12:             **end if**
13:         **end if**
14:     **end for**
15:     **return** $tlus$
16: **end procedure**

---

| Model | ARC | SCONE | GSM8K | AQUA-RAT | AR-LSAT |
|---|---|---|---|---|---|
| **Answer Accuracy** | | | | | |
| Random Guess | 25.0 | — | — | 20.0 | 20.0 |
| GPT-3 [`davinci`] (few-shot) | 72.9 | 02.3 | 34.8 | 40.2 | 19.0 |
| GPT-3 [`davinci`] (few-shot, MT) | 88.1 | 01.2 | 07.0 | 37.1 | 20.0 |
| GPT-3 [`curie`] (few-shot) | 29.4 | 00.0 | 0.37 | 24.8 | 21.0 |
| **Graph Similarity** | | | | | |
| GPT-3 [`davinci`] (few-shot) | 15.1 | 01.9 | 16.0 | 05.2 | 01.1 |
| GPT-3 [`davinci`] (few-shot, MT) | 25.3 | 01.2 | 0.21 | 05.8 | 00.9 |
| GPT-3 [`curie`] (few-shot) | 00.9 | 00.0 | 00.0 | 00.3 | 00.0 |

Table 4: Results on the test set across the different tasks and different evaluation metrics for variations of the GPT-3 model. Numbers are in percentage. The "Random Guess" results are included to facilitate visualization since different tasks have different answer types.

## C    IMPLEMENTATION DETAILS

The T5 model is fine-tuned in an auto-regressive manner. We select the AdamW (Loshchilov & Hutter, 2019) as the optimizer. During training, we use batches containing two data points. The learning rate starts at zero and is gradually increased to its maximum value of $3 * 10^{-5}$. After 1000 steps, the learning rate is decreased following a cosine function scheduler. The weight decay is set to $10^{-3}$.

## D    REASONING GRAPH SIMILARITY

The textual similarity function $\sigma(a, a)$ is a binary function that maps the text of two nodes ($a$ and $b$) to a TRUE or FALSE value. We use different textual similarity functions for each of the tasks in STREET. The description of the function are found in Table 5.

Note that our comparison function disregard reasoning steps that contain no antecedents as they are mostly used as supporting facts during reasoning (for instance, "one hour has 60 minutes" is a node that might not have antecedents in the reasoning graph).

| STREET Tasks | Text Similarity $\sigma(a, b)$ Description |
|---|---|
| SCONE | The nodes in scone follow a well defined textual structure, therefore the node similarity returns TRUE if and only if $a = b$. |
| GSM8K and AQUA-RAT | We parse the node text and extract all the mathematical values inside the node. For instance "Natalia sold 48 / 2 = 24 clips in May" would be converted to the set $\{48, 2, 24\}$. Then the similarity function return TRUE if the sets extracted from $a$ and $b$ are equal. |
| ARC and AR-LSAT | Since the conclusion nodes in these are in free text format, we follow Dalvi et al. (2021) and use the $BLEURT$ (Sellam et al., 2020) text similarity function. $BLEURT$ is a *trained metric* that uses the BERT language model and was shown to have better correlation to human judgment than standard metrics like $BLEU$ and $ROUGE$. We define that $BLEURT > 0.25$ is the threshold that decides if nodes $a$ and $b$ are similar. |

Table 5: Definition of the text similarity function for each of the STREET tasks.

