# OpenReview forum: "STREET: A MULTI-TASK STRUCTURED REASONING AND EXPLANATION BENCHMARK"
_ICLR.cc/2023/Conference — ICLR 2023 notable top 25%_

### Official Review · Reviewer_KKP2 · 2022-10-14

**Confidence:** 4
**Correctness:** 4
**Technical Novelty And Significance:** 3
**Empirical Novelty And Significance:** 3
**Recommendation:** 8

**Clarity, Quality, Novelty And Reproducibility:**

The paper is clearly written and easy to follow. The proposed work is novel and very interesting.
Below are some clarification questions for the authors:

**Clarification Question (CQ1)**: In the text example shown in Section 3.1, why do you need to have (3) in the proof steps that does (1)+(3)+(4)->(5) ? (1) and (4) should give enough information to infer (5): 72 clips in April and May.

**Clarification Question (CQ2)**:  In the Appendix A.1, in the SCONE (Alchemy) example, how is the agent supposed to know that mixing 1 yellow and 1 green chemical gives 2 brown chemicals? Same question for (17)+(12)->(19).

**Clarification Question (CQ3)**: In the Appendix A.1, in the GSM8k example I believe conclusion (8) should have incoming edge (6) in addition to (7). If not, can you explain why the proof has a (5) -> (6) step since (6) is not used anywhere in the proof.

**Recommendation (R3)**: This makes me think about another statistic that would be interesting to have: how many proof steps in the graph are not used to arrive at the final answer? These can be identified as leaf nodes that are not the answer in your graph.

**Strength And Weaknesses:**

**Strengths**:
The paper is clear and easy to read. The motivation of the work is well defined.
The introduction of an additional multi-step reasoning benchmark is very welcomed and can have a large impact as many previous multi-hop datasets were either only a few hops (mainly 2), or lacking the natural language steps to explain complex answers.

**Weakness (W1)**: However, some previous datasets propose multi-step reasonings with logical explanations in natural language: CLUTRR (Sinha et. al, 2019) and AI2 ProofWriter (Tafjord et. al, 2020). The paper must compare their benchmark to these previous work.

**Proposition (P1)**: It would be nice to include CLUTRR and ProofWriter in the STREET benchmark or at least document how the community can contribute to STREET and add new relevant multi-step reasoning datasets in time.

**Proposition (P2)**: Can the authors compare models’ performance when the candidate answers are present and missing from the input to the model.

**Recommendation (R1)**: The STREET benchmark should also propose a split based on the number of reasoning steps per dataset. That would give users the opportunity to evaluate (1) a multi-task setup, and (2) a compositional generalization setup: can a model trained on k reasoning steps perform well on k+-n reasoning steps.

**Question (Q1)**: The reasoning graph’s steps are ordered according to a valid topological order. However, there can exist multiple topological orders. For instance if we have A+B->D followed by B+C->E, followed by D+E->Final Answer, we can also first prove B+C->E then A+B->D then E+D->Final Answer. When linearizing the graph into a sequence of tokens, did the authors shuffle the premise order? Otherwise trained models could pick on some weird premise order biases. Something to consider.

**Recommendation (R2)**: The text similarity function for GSM8K and AQUA-RAT should probably include the operator characters. If a reasoning step contains 48*2=24 and the golden step was 48/2=24 the current evaluation will consider the predicted step correct.

**Weakness (W2)**: The paper should include previous best performance on each individual dataset. This can be done by adding a “best model” line to the Answer Accuracy section of Table2, and detailing in the text of the paper or the caption of the table which ‘best’ model refers to for each of the dataset.



**Summary Of The Paper:**

This work introduces a new benchmark (STREET) to evaluate the reasoning capacity of language models. The benchmark contains 5 different reasoning datasets (ARC, SCONE, GSM8K, AQUA-RAT, AR-LSAT) requiring on average 7.8 reasoning steps per answer.
This benchmark evaluates 3 metrics: Answer Accuracy, Reasoning Graph Accuracy, and Reasoning Graph Similarity.
The paper evaluates a t5-large model fine-tuned on individual tasks and a GPT3 model in a few-shot procedure. Results show that large language models still struggle in these multi-step reasoning tasks.


**Summary Of The Review:**

This is a good paper, however some things need to be added: see **W1**, **W2**, and **R1**, **R2**, **R3** paragraphs in the above sections. In addition, after looking at the few examples of the dataset in the paper, I question the quality of the reasoning graphs automatically generated (see **CQ1**, **CQ2** and **CQ3** above.)

---

> ### Author Response · Authors · 2022-11-19
> **Response to Reviewer KKP2**
>
> Answer to weakness:
>
> W1 - Thank you for sharing these related works. We mentioned AI2 ProofWriter with the RuleTaker D* dataset (Tafjord et. al, 2020) in the previous version, but we added further comparisons of these two datasets in section 5.
>
> W2 - Thank you for the suggestion, but we did not include the published SOTA results to Table 4 since it would *not* be a fair comparison (the STREET test set is different from original datasets. We used a subset of the original data due to annotation costs. Furthermore, some of the original questions were modified and now include reasoning premises). We will defer to the readers to find SOTA results of the original datasets, or we can include the results in the Appendix on the camera-ready version.
>
> Answer to recommendations & propositions:
>
> P1 - The reasons for not including CLUTRR and ProofWriter are 1) most of their data is semi-synthetic and 2) it is relatively simple for large generative models to solve (for instance, published results of ProofWriter show ~99% ACC for answers and ~97% ACC for proof generation). We focused mostly on harder tasks with less synthetic data.
>
> P2 - This would be difficult to do for certain tasks (e.g., AR-LSAT needs multiple-choice options for some questions like "which of the following cannot be true?"). We wanted to keep the same QA formulation to avoid issues with introducing unsolvable questions, or making questions too hard/easy.
>
> R1 - We can mention "split based on the number of reasoning steps per dataset" in the future work section of our paper.
>
> R2 - That's a good point, we will modify the evaluation code and update the results for the camera-ready version.
>
> R3 - Thank you for the suggestion, we will add this statistic to the camera-ready version.
>
> Answer to questions:
>
> Q1 - We select the topological order with lower node numbers appearing first. However, trying to circumvent the problem of models "picking up premise order biases" definitely sounds like a possible avenue for future research.
>
> CQ1 - That was a design choice. We wanted to make sure that the question is always present as a node in the graph, so in this example the node (3) was indeed optional, but useful to know what we are answering the question on node (5).
>
> CQ2 - The SCONE dataset arbitrarily defines that mixing two chemicals of different colors will always create a brown chemical, so the model has to learn that from the examples.
>
> CQ3 - You are correct, that is a missing edge.

---

### Official Review · Reviewer_xxyE · 2022-10-24

**Confidence:** 3
**Correctness:** 3
**Technical Novelty And Significance:** 2
**Empirical Novelty And Significance:** 2
**Recommendation:** 5

**Clarity, Quality, Novelty And Reproducibility:**

The paper has good clarity and quality.
The authors promise to release data, code and a leaderboard, so reproducibility is good.

**Strength And Weaknesses:**

Strength:
* The paper studies an important problem: reasoning capabilities when answering complex questions.
* The STREET benchmark, especially the reasoning graphs that are newly annotated, is a useful resource for studying structured reasoning and explanations. (Thanks to the authors for these huge efforts!)
* Paper is written clearly and is easy to follow.

Weaknesses:
* Lack of comparisons and justifications
  * Since multi-tasking and a unified reasoning formulation is one of the main contribution, it is necessary to include non-multitask performance of the five datasets in order to justify whether multi-tasking is useful.
  * To justify the proposed unified reasoning formulation is useful, it is necessary to include a multi-task baseline with no reasoning and proofs.
  * To justify that the proposed reasoning graph is more flexible, it is necessary to compare performance (on ARC, where annotations are available for both methods) of reasoning graph and entailment tree. Also please consider including examples that entailment trees are not able to represent but reasoning graphs are.
* Findings are mostly comparing performance on the surface-level.
  * In sec 4.2.1, error analysis is conducted on each dataset _separately_. As a multi-task benchmark, it is important to show what is lacking from current best methods with a holistic view, and what are the potential directions for future research.
  * I will be very interested in a case study on reasoning graphs generated by the models. How often do they generate a valid reasoning graph in the output?
  * I will also be very interested in quantifying performance by grouping test set with reasoning steps. Does the model struggle more on test cases with more reasoning steps?
* More details needed on annotation process
  * Since the resource of reasoning graphs is one main contribution of the paper, it is important to provide more information about how expert annotators are selected, whether inter-annotator agreement is measured (as mentioned, "there are multiple ways one can explain the answer").
  * This is less important but please provide more information on the user-interface mentioned in Sec 2.3.

Questions:
I have some questions regarding some choices in STREET.
* In Table 3, different similarity function is used for different dataset. I suspect that extracting and comparing numbers or using BLEURT > 0.25 may be over-simplified, but I may be wrong. Do you observe any false negative/positives? How often does this happen?
* Is there any reason that an adapted version of HotpotQA (e.g., using the gold context sentences, so that information needed to answer the question is present) is not included in STREET.


**Summary Of The Paper:**

The paper presents STREET, a multi-task benchmark for structured reasoning and explanations in NLP.
* STREET is composed of 5 existing datasets: ARC, SCONE, GSM8K, AQUA-RAT and AR-LSAT.
* The authors introduced a unified reasoning formulation involving textual logical units (TLUs) and reasoning graphs.
* The authors used expert annotation or automatic programatic annotation to annotate the reasoning process on a subset of the 5 datasets mentioned above.
* The authors introduce three metrics (answer accuracy, reasoning graph accuracy, and reasoning graph similarity) to evaluate a system trained on STREET.

Empirically, the authors evaluate T5-large fine-tuning and GPT-3 in-context few-shot learning performance on STREET, and explain errors on a per-dataset basis.

**Summary Of The Review:**

The reasoning formalism and resources introduced in the paper will be useful to the research community.
However, the paper could benefit from more detailed empirical analysis on the proposed benchmark.
Also, some experiment settings and designs need to be further justified or explained (e.g., what's the motivation of multi-tasking?)

---

> ### Author Response · Authors · 2022-11-19
> **Response to Reviewer xxyE**
>
> Answer to weakness:
>
> 1.1 - Thank you for the suggestion. We included the results simulating a multi-task setting with GPT-3 (Appendix B). We also plan to add results with T5 models trained on all tasks simultaneously in the camera-ready version.
>
> 1.2 - Our understanding is that "multi-task baseline with no reasoning and proofs" is a baseline that only answers the questions. However, standard QA is outside the scope of our paper, which focuses on explanation and multi-step reasoning.
>
> 1.3 - We point out the differences between "reasoning graphs" and "entailment trees" in section 5, we also show two examples in figure 1 where the reasoning graphs contain nodes with more than one parent (which can't be represented as tree structures, especially in the case of proof with larger "depth"). We added comments further explaining this difference (Section 2.1).
>
> 2.1 - Thank you for your suggestion. We plan to add an error analysis of all tasks together, as well as possible avenues of research in the camera-ready version of the paper.
>
> 2.2 - In section 5 we try to capture "How often do they generate a valid reasoning graph" by using the "Reasoning Graph Similarity" metric, we also perform a comparison against human performance in Section 4.2
>
> 2.3 - Thank you for your suggestion. We plan to add a graph with metrics broken down by the number of reasoning steps for all models in the camera-ready version of the paper.
>
> 3.1 and 3.2 - We further explain the annotation process and include: details about the annotation process and instructions given to workers (Appendix A.2), the education level of annotators (section 2.3), as well as a description of the User Interface, with images (Appendix A.2).
>
> Answer to questions:
>
> 1 - There can certainly be false positive/negative examples of nodes, but we found that the metric correlates with human generation. The BLURT scores were also used by Dalvi et al. (2021) which contain an analysis showing high correlation with human-scored examples.
>
> 2 - HotpotQA explanations are relatively shallow, containing an average of 2 to 3 facts / edges per explanation (far below what we wanted to have in STREET, which is multi-step and multi-premise). Furthermore, most of the explanations do not follow any explicit order of inference, so the structured version of this dataset would not be very relevant.

---

### Official Review · Reviewer_3BMb · 2022-10-25

**Confidence:** 3
**Correctness:** 3
**Technical Novelty And Significance:** 3
**Empirical Novelty And Significance:** 3
**Recommendation:** 8

**Clarity, Quality, Novelty And Reproducibility:**

The paper is clear, and there is strong novelty in defining a
challenging and useful task.  The dataset will be available, and the
techniques used can be reproduced.


**Strength And Weaknesses:**

Strengths:

-- A challenging task with much annotations in several domains

-- An analysis of performance of several advanced techniques

Weaknesses:

-- A few minor clarity issues

typos, etc:

Page3: "The TLUs for the rationale L_R can be thought of .. " is
confusing ..  (I don't think L_R is defined yet, though L_T is, and
perhaps you meant to say: 'The TLUs for the rationale R, L_R, can be
...", that is introduce the notation L_R pehraps ..) (L_R is used in
the next paragraph.. ) (On re-reading, I now understand T can be any of
L, O, etc.. perhaps stress that, early, when you start with for any T
in \bf{T}..  )

page2: why not use 'correct' instead of 'expected' answer.

A couple of questions:

question 1: [Assessing the quality of graph matching] Especially when
the system gets the answer right, and the graph generated is perhaps
somewhat similar but not perfect, how often are the steps actually
correct or close to correct (is it near 0%?).  It would be good to
understand the calibration of your graph matching, or to get a
function mapping from similarity score to correctness or degree of
connectedness, by manually examining a few such cases, perhaps 10s to
100s is sufficient to get an estimate (you may have done this
analysis, but I missed it) (I looked at appendix C but didnt find
such). It is possible that a partial graph similarity score is almost
never correct (with probability 90% or 99% depending on how many you
manually evaluate).

question 2: Is there any need for negation (does that happen, or is
the percentage of such basically 0)?  For instance, a part of the
question or context negating one of the multiple choice answers,
thereby rendering it incorrect.



**Summary Of The Paper:**


The authors define a reasoning task, that of providing a graph that
corresponds to the reasoning steps leading to the correct answer (from
premises provided in the question), and build an annotated dataset
with question and answers and the graphs from several domains. The
authors present several properties of the dataset (histograms of
number of steps required, number of premises, etc). They explain how they use existing
techniques to produce their reasoning, how they evaluate (graph
comparisons), and report that existing techniques appear to
substantially lag behind human performance (even though, the correct
answer rate may be relatively high).



**Summary Of The Review:**


A useful challenging task that will help advance language understanding and research on reasoning/explanations.

---

> ### Author Response · Authors · 2022-11-19
> **Response to Reviewer 3BMb**
>
> Answer to questions:
>
> 1 - Figure 3 in section 4.2 attempts to assess the quality of the graph matching metric (specifically for correct answers given by both humans and baseline models). Since this is a relevant question, we plan to perform a manual evaluation of outputs and summarize our findings in the camera-ready version of the paper.
>
> 2 - Negation is fairly important in our tasks, but all negations are handled in natural language (e.g., the AR-LSAT explanation in Figure 1 contains "(16) If new jazz or used jazz are NOT on sale [...]" or "(18) If NEITHER type of jazz is on sale, both types of [...]"). Due to this reason, we chose not to explicitly annotate the nodes or edges with any kind of "negation label".

---

### Official Review · Reviewer_4X1J · 2022-10-27

**Confidence:** 4
**Clarity, Quality, Novelty And Reproducibility:** Clarity is good, but Quality, Novelty…
**Correctness:** 3
**Technical Novelty And Significance:** 3
**Empirical Novelty And Significance:** 3
**Recommendation:** 6

**Strength And Weaknesses:**

Strengths:
1. explainable NLP is an active research area, and this benchmark can help the community push forward the progress
2. Some baseline results are estalished

Weaknesses:
1. There is a lack of detail in annotation process.  Specifically, AR-LSAT seems like an extremely difficult reasoning dataset, and yet in section 2.2, very limited information about how the annotation process is carried out.
2. There is a lack of dataset quality verification process / dataset analysis.  There are many papers showing that the agreement rate could be low for intermediate reasoning steps annotations.  An in-depth study in this regard is important.  Examples include sec 3.4 in [1] and and sec 4 [2].
3. As a multi-task dataset, there is a lack of analysis in the multi-task aspect.  At the bare minimum, it would nice to see how fine-tuning on all of the tasks improves or degrades the performance. See [3] for an example of what people do to analyze multi-task models.

Questions / suggestions:
1. What is the rationale of not including previous multi-hop reasoning dataset such as ProofWriter?  Are the authors only selecting datasets that contain pure natural language but not synthetically generated language?
2. It would be nice if the authors could discuss what more can this paper offer than [4], which has a more comprehensive survey on the structured explanation dataset.  I guess the unifying metrics established in this benchmark is valuable for future work to evaluate different NLP systems under the multi-task setting.

[1] WinoWhy: A Deep Diagnosis of Essential Commonsense Knowledge for Answering Winograd Schema Challenge
[2] HOTPOTQA: A Dataset for Diverse, Explainable Multi-hop Question Answering
[3] UNICORN on RAINBOW: A Universal Commonsense Reasoning Model on a New Multitask Benchmark
[4] Teach Me to Explain: A Review of Datasets for Explainable NLP

**Summary Of The Paper:**

1. This paper presents STREET, a new question answering benchmark that also contains step-by-step structured explanation annotations.
2. Established a number of baselines on this dataset including GPT3 few-shot prompting.

**Summary Of The Review:**

This paper presents a valuable benchmark for pushing forward research in structured explainability, and I would love to see it accepted in the end.  In the current version, 1) authors spent most effort in collecting annotations alone and have done little analysis for the annotations, and 2) although being a multi-task benchmark, this is a lack of analysis in the multi-task aspect because experiments have mostly been done on each task individually. For these two reasons, I don't feel this paper is ready to be presented.

---

> ### Author Response · Authors · 2022-11-19
> **Response to Reviewer 4X1J**
>
> Comments on weakness:
>
> 1 - We believe Section 2 contains a non-trivial amount of annotation details, but we further explain the annotation process by adding to the revised paper: details about the annotation process and instructions given to workers (Appendix A.2), the education level of annotators (section 2.3), as well as a description of the User Interface, with images (Appendix A.2).
>
> 2 - We have added an inter annotation agreement, with metrics suggesting that the data has good quality (Section 2.3) and further dataset analysis (Appendix A.4).
>
> 3 - We included the results simulating a multi-task setting with GPT-3 (Appendix B). We also plan to add results with T5 models trained on all tasks simultaneously in the camera-ready version.
>
> Answer to questions:
>
> 1 - The reasons for not including the ProofWriter are 1) some of its data is purely synthetic and 2) it is relatively simple for large generative models to solve (published results are ~99% for answers and ~97% for proof generation).
>
> 2 - All the datasets from [4] are either non-structured (e.g., NATURAL-QUESTIONS, E-SNLI, WINOWHY) or contain structured explanations, but have few "hops" (at most 3)  and contain "shallow structure" (represented as a list of consecutive hops, instead of a graph) (e.g., EQASC, STRATEGYQA, OPENBOOKQA). Our proposed STREET dataset goes beyond the simple "multi-hop" setting, creating a dataset from challenging reasoning QA problems and adding rich, multi-step, multi-premise, and structured explanations.

---

> > ### Comment · Reviewer_4X1J · 2022-11-21
> > **Revision is good**
> >
> > Thanks to the authors for adding the annotation detail.  It is really helpful.  I have updated the paper score to reflect the changes in the revision.
> >
> > FWIW, I still think there is a lack of literature review in structured explanations.  For example, "ExplaGraphs: An Explanation Graph Generation Task for Structured Commonsense Reasoning" and "MuSiQue: Multihop Questions via Single-hop Question Composition" both collect structured explanations.
> >
> > Finally, I would also urge the authors to address the feedback posed by reviewer xxyE.
> >
> > Otherwise, I have no objections seeing this paper accepted and believe this paper can bring significant value to the community for building interpretable, robust NLP systems.

---

> > > ### Author Response · Authors · 2022-11-23
> > > **Response #2 to Reviewer 4X1J**
> > >
> > > Thank you for your valuable feedback and for updating the paper score.
> > >
> > > We will add both papers to the related work section and will further address the feedback from reviewer xxyE (details in their the individual response).

---

### Author Response · Authors · 2022-11-19
**Response to all reviewers**

We thank all the reviewers for their valuable and thoughtful comments and suggestions. We've uploaded a revised version of the paper, which has addressed most of the concerns. A summary of the major changes are listed below:

1 - We added further annotation details, including instructions given to annotators and image with annotation tool

2 - We computed inter annotation agreement (Fleiss' kappa)

3 - We include a dataset analysis, showing the different types of reasoning inferences in the dataset

4 - We added further results, using different model sizes and simulating the multi-task setting with GPT-3

We also include responses to each individual reviewer, answering questions and suggestions. We are hopeful that our responses have addressed your concerns.

---

### Decision · Program_Chairs · 2023-01-20

**Decision:**

Accept: notable-top-25%

**Justification For Why Not Higher Score:**

While this paper proposes a benchmark for an important problem, I believe that the longer presentation and the more extensive reach that an oral would give are not justified, since the benchmark can be summarised quickly and the problem is still quite niche in the wider research community.

**Justification For Why Not Lower Score:**

Getting foundation models better at logical reasoning is an important problem that justified extra attention from the community. The paper also received favourable reviews, so I believe a spotlight is justified.

**Metareview: Summary, Strengths And Weaknesses:**

This paper proposes a new benchmark of datasets that require multi-step reasoning to solve them. This paper aggregates 5 existing datasets (ARC, SCONE, GSM8K, AQUA-RAT and AR-LSAT) into a common benchmark with a unified reasoning formulation based on textual logical units and reasoning graphs. Where necessary, the authors did extra annotation to achieve this format. The authors then introduce three metrics (answer accuracy, reasoning graph accuracy, and reasoning graph similarity) to evaluate GPT-3 and T5 on this benchmark.

Logical reasoning is a known shortcoming of large language models that is important to address by the community. This benchmark should be a good stepping stone to make progress in this direction.

The reviewers suggested that this paper can be further improved by splitting the datasets into subsets based on the number of reasoning steps required to solve each questions and reporting results on these subsets. The reviewers also suggested adding an analyses of the types of errors that the models make on these problems to highlights research questions for the community. The authors promised to include these in the camera ready version of the paper, so I am happy to accept it.


**Note From Pc:**

if the above contains the word "oral" or "spotlight" please see: "oral" presentation means -> notable-top-5% and "spotlight" means -> notable-top-25%. As stated in our emails, we are disassociating presentation type from AC recommendations